# Mechanisms of school-based peer education interventions to improve young people's health literacy or health behaviours: A realist-informed systematic review

Emily Widnall[1]*, Steven Dodd[2], Abigail Emma Russell[3], Esther Curtin[1], Ruth Simmonds[4], Mark Limmer[2], Judi Kidger[1]

1 Centre for Public Health, University of Bristol, Bristol, United Kingdom, 2 Faculty of Health and Medicine, Lancaster University, Lancaster, United Kingdom, 3 College of Medicine and Health, University of Exeter, Exeter, United Kingdom, 4 Mental Health Foundation, London, United Kingdom

* emily.widnall@bristol.ac.uk

**Data Availability Statement:** All relevant data are within the paper and its Supporting Information files.

## Abstract

### Introduction

Peer education interventions are widely used in secondary schools with an aim to improve students' health literacy and/or health behaviours. Although peer education is a popular intervention technique with some evidence of effectiveness, we know relatively little about the key components that lead to health improvements among young people, or components that may be less helpful. This review aims to identify the main mechanisms involved in school-based peer education health interventions for 11–18-year-olds.

### Methods

Five electronic databases were searched for eligible studies during October 2020, an updated search was then conducted in January 2023 to incorporate any new studies published between November 2020 and January 2023. To be included in the review, studies must have evaluated a school-based peer education intervention designed to address aspects of the health of students aged 11-18 years old and contain data relevant to mechanisms of effect of these interventions. No restrictions were placed on publication date, or country but only manuscripts available in English language were included.

### Results

Forty papers were identified for inclusion with a total of 116 references to intervention mechanisms which were subsequently grouped thematically into 10 key mechanisms. The four most common mechanisms discussed were: 1) Peerness; similar, relatable and credible 2) A balance between autonomy and support, 3) School values and broader change in school culture; and 4) Informal, innovative and personalised delivery methods. Mechanisms were identified in quantitative, qualitative and mixed methods intervention evaluations.

**Funding:** This study was funded by the National Institute for Health and Care Research (NIHR) School for Public Health Research (project number SPHR PHPES025). The views presented in this study are those of the author(s) and not necessarily those of the NIHR or the Department of Health and Social Care. The funders had no role in study design, data collection and analysis, decision to publish, or preparation of the manuscript.

**Competing interests:** The authors have declared that no competing interests exist.

## Discussion

This study highlights a number of key mechanisms that can be used to inform development of future school-based peer education health interventions to maximise effectiveness. Future studies should aim to create theories of change or logic models, and then test the key mechanisms, rather than relying on untested theoretical assumptions. Future work should also examine whether particular mechanisms may lead to harm, and also whether certain mechanisms are more or less important to address different health issues, or whether a set of generic mechanisms always need to be activated for success.

## Introduction

School-based health interventions offer a valuable opportunity for prevention and early intervention to ensure good health and well-being in school-aged children. One key area of intervention is improving health literacy and health behaviours among adolescents which is an important public health topic given the strong links between health literacy and adult health outcomes [1] as well as generally promoting health during the life course [2]. One popular approach is the use of peer-to-peer teaching methods, with evidence of a global rise in peer education interventions over the past few decades [3, 4]. The literature offers several reasons as to why peer education interventions are popular and likely to be an effective method for improving health literacy and health behaviours. The importance of peer influence, particularly in adolescence, is well-documented [5, 6], and there is evidence that young people are more likely to seek help for health and well-being from informal sources of support such as friends in comparison to a school teacher or trusted adults [7]. Peers also play an important role within their school community and are likely to be seen as more relatable than teachers with less of an imbalance of authority [8]. Peers have also been found to be seen as role models by younger students in previous research [9].

Many assumptions on the effectiveness of peer education for health improvement centre around adolescence. For example, discussing adolescence as a key period for establishing health-related beliefs and opinions and decision-making in relation to health behaviours [10], as well as discussion of an increase in risk-taking behaviour in adolescence [11]. Many also argue that peer education is based on the rationale that peers have a stronger influence on individual behaviour due to the level of familiarity and trust and the comfort they are able to provide [12].

Existing peer education interventions cover a wide range of health areas, including mental health, physical health, sexual health, and health promotion and behaviour (e.g., healthy eating and smoking prevention [13–16]). This review focuses specifically on peer education, typically involving the selection and training of 'peer educators', who subsequently teach or support younger or similar aged students in their school, known as 'peer learners'. Although there are variations in peer-led interventions including 1:1 peer mentoring and counselling [17–20], these are beyond the scope of this review.

A number of key theories have been suggested as underpinning peer education. The most widely cited key theoretical underpinnings within the peer education literature include Bandura's Social Learning/Social Cognitive Theory [21, 22]; the Diffusion of Innovation Theory [23, 24] and within the peer education health literature, the Health Belief Model [25].

Although key theories are often stated within peer education papers, studies often fail to show clearly how the outcomes are derived from theoretical constructs [26]. Despite the range

of theoretical approaches discussed within the peer education literature, there remains a lack of evidence for the specific mechanisms at play which lead to improved health outcomes in peer-led interventions. There has also been a call for the development of logic models to identify the change mechanisms that lead to changes in health literacy and/or behaviours [27].

Traditional systematic review approaches have often been criticised for being overly specific and rigid [28–30]. Although a number of reviews exist to assess the effectiveness of peer education interventions, they typically lack in explaining why these interventions may or may not work, in what contexts, and under what circumstances. Realist reviews have emerged as a strategy for synthesising evidence for complex social interventions, such as for peer education [31–33] and other complex health-related interventions such as housing and mental health programs [34–36]. Realist reviews provide an explanatory focus to understand and unpack the mechanisms by which an intervention works, for whom, and in what circumstances [29].

The aim of this review is to address the limitations in the existing literature, in particular the lack of current exploration of mechanisms of change, by identifying the key mechanisms involved in school-based peer education interventions that aim to improve health literacy and health behaviours in young people.

## Methods

The PICO (Population, Intervention, Comparator and Outcome) format was followed when developing our research questions and this review was completed in accordance with the 2009 PRISMA statement [37] and pre-registered on PROSPERO (CRD42021229192). A Prisma checklist can be found in S1 Table.

### Realist approach to systematic review

Rather than focusing on meta-analysis and pooling intervention effect sizes, realist reviews seek to generate learning and insights into why interventions work or do not work and what explains these effects [29]. Additionally, a realist review uses the contextual characteristics of programs to help explain program successes and/or failures. This approach to evaluating existing evidence is explanatory because it combines both theoretical thinking and empirical evidence about how and why interventions work (or do not work).

### Search strategy and selection criteria

The search process was part of a wider review of effectiveness of school-based peer education interventions to improve young people's health [38]. Although the initial search identified a larger number of studies, this review only reports on 40 papers that directly comment on mechanisms of change involved in the peer education interventions. Initial searches as part of the wider review of effectiveness were conducted during October 2020. An updated search was then conducted in January 2023 to incorporate any new studies published between November 2020 and January 2023 that contained data on mechanisms of effect.

Five electronic databases were searched for eligible studies: CINAHL, Embase, ERIC, MEDLINE and PsycInfo. Search terms were developed by looking at key texts and in discussion with the study team and involved pilot searches and subsequent refinements. An example of the search terms can be found in the S2 Table. Given the lack of literature on peer education mechanisms and the exploratory nature of this review, no restrictions were placed on publication date, country or language.

The inclusion and exclusion criteria for the review are included in Table 1.

**Table 1. Inclusion and exclusion criteria.**

| *Inclusion Criteria* | *Exclusion Criteria* |
|---|---|
| 1) Studies must evaluate a peer education intervention where students take a lead in the education of other students. | 1) Peer education interventions concerning academic outcomes (e.g., reading and writing achievement). |
| 2) Both peer educators and peer learners involved in the intervention must be aged between 11-18. | 2) Interventions concerning anger management, behavioural problems, or social skills. |
| 3) The intervention must target health-related outcomes (health literacy, or health behaviours). | 3) Interventions concerning traffic safety, health and safety, avoidance of injuries, or first aid. |
| 4) The interventions must have taken place within a school and during school hours. | 4) Interventions concerning cultural, social, or political awareness (e.g., media literacy). |
| 5) The interventions must have involved universal peer education in which whole classes or year groups are the target of the intervention. | 5) Interventions in which health outcomes are secondary to other outcomes (e.g., interventions focused on reading that indirectly improve self-esteem). |
| 6) Studies must contain primary qualitative and/or quantitative data and report data or themes pertaining to mechanisms of school-based peer education health interventions. For quantitative papers, studies had to report data that may explain why an outcome was seen and provide a reflection on potential mechanisms at play. This could also include discussing numerical outcome data in line with existing theoretical underpinnings of peer education. | 6) Interventions that were not universal in nature and instead focused on specific at-risk groups. |
| | 7) One-to-one mentoring interventions. |
| | 8) Do not make specific reference to mechanisms of effectiveness/ineffectiveness. |

## Primary outcome(s)

1) Evidence of mechanisms or factors associated with peer education interventions that explain why improvements were or were not seen in participant health.

**Data extraction, selection and coding.** Two authors (SD and EW) independently screened papers according to the inclusion criteria above using the Rayyan online review platform (https://www.rayyan.ai/). Any cases of uncertainty or disagreement were discussed and agreed among the wider research team.

Two authors (SD and EW) independently extracted the data, discussing and resolving any discrepancies that arose. Typically, disagreements arose during the initial coding process where we detailing types of mechanisms appearing in the paper, before we had consolidated the themes and confirmed the names of the mechanisms. These discrepancies were resolved through iterative discussions between EW and SD as well as with the wider team.

Data extraction included author, year of publication, location, study aim, design and sample size, description of the intervention, outcome measures and findings relating to mechanisms of peer education interventions. All data relating to how and why the peer education intervention was thought to be effective (or ineffective) was extracted onto a spreadsheet.

In deciding whether data or themes were pertinent to the synthesis, reviewers considered if the identified data offered an explanatory account of what was going on between the intervention(s) and its outcomes. For qualitative studies, data were derived from themes, participant quotes and lesson observations. For quantitative studies, data were derived from author reflections on intervention content and reflections on numerical data as insight into underlying mechanisms.

EW and SD conducted a thematic analysis of data about reasons for effectiveness to identify overarching themes and create a table of 'key mechanisms' (Table 3). The analysis followed some aspects of the framework approach [39], primarily by creating an analytical framework to code all extracted mechanisms. After reading the included papers and extracting any relevant mechanism data into the extraction table, EW and SD made initial notes and started a set of preliminary codes. EW and SD agreed the list of codes which closely described all the mechanisms that were discussed in the included papers. This process went through several iterations through discussions between EW and SD as well as the wider author team. Once the coding framework (set of identified mechanisms) was agreed upon, EW and SD used the final framework to code all relevant mechanisms data within the extraction table. We went through an iterative process of comparing and consolidating mechanisms between papers and reduced an initial larger number of mechanisms to reach the final 10 included.

## Quality appraisal

The Mixed Methods Appraisal Tool (MMAT) was used to assess quality of reporting procedures. This tool consists of five specific quality rating items depending on study design (qualitative, quantitative randomized, quantitative non-randomized, quantitative descriptive and quantitative mixed methods). Each paper was given a rating from 0-5 using the relevant questions depending on study design. The following ratings were used to summarise study quality: 0-1 indicating poor quality, 2-3 indicating average quality and 4-5 indicating high quality. Quality ratings of all included papers can be found in S3 Table.

## Results

A total of 2,474 studies were identified after the searches and 40 studies were eligible for inclusion. Fig 1 illustrates a flow diagram of the search. A summary of all included studies can be found in S3 *Table* which includes study author, location and year, health area under investigation, sample size, study design, summary of key mechanisms, how the mechanisms were identified as well as the quality score and rating. The table also records whether the intervention under evaluation used a logic model and whether the paper specifically referred to mechanisms of change.

Of the 40 papers included in this review, only two papers referenced a logic model [40, 41] and only 9 papers specifically referred to the word 'mechanism(s)' within their write-up, and this tended to be in the context of calling for future research to focus on intervention mechanisms that lead to effectiveness.

Of the 40 included studies, 20 were mixed methods evaluations, 15 were quantitative studies and 5 were qualitative studies. Typically, mechanisms discussed in relation to quantitative outcomes were in line with key theoretical underpinnings and through authors' reflections on outcome data, whereas in mixed methods and qualitative papers, mechanisms were identified directly from qualitative themes or quotes. The findings in these studies were derived from the views of young people and teachers in interview/focus group data, classroom observations, and pre and post intervention self-report surveys.

Table 2 details all health areas and study designs of included studies. The specific health area of interest in each study is also detailed in S3 Table.

Data extraction identified 116 mechanisms referenced within the 40 papers, which were categorised thematically into 10 'key mechanism' groupings (see Table 3). Studies typically discussed 2-3 mechanisms each, but papers ranged from discussing one to six individual mechanisms.

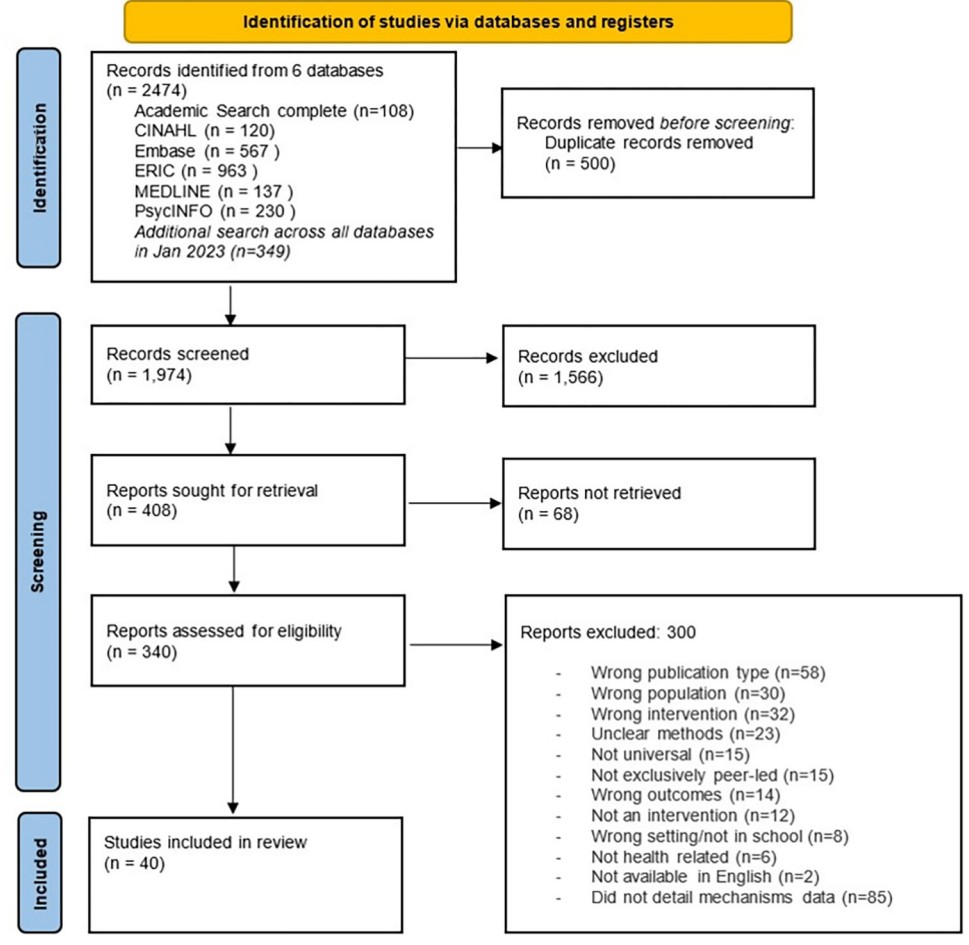

**Fig 1. Full search strategy.**

Study quality varied with over one-third of studies (n = 15) rated as high quality, half of the studies rated as medium quality (n = 20) and five studies rated as low quality (13.8%). Several studies lacked detailed description of methodology and many had incomplete outcome data.

Mechanisms discussed in relation to quantitative study findings often centred around existing theoretical assumptions. Table 4 lists the theories cited within the 35 included papers. Theories typically centred around interpersonal influences, social reinforcement and peers acting as role models. The most widely cited theories were Bandura's Social Learning/Cognitive Theory [21, 22] and Diffusion of Innovation Theory [23].

**Table 2. Number of included papers by health area.**

| Health area | Included papers | Quantitative | Qualitative | Mixed methods | Number of mechanisms references |
|---|---|---|---|---|---|
| Alcohol, smoking, substance use | 13 | 7 | 3 | 3 | 38 |
| Sex education | 13 | 2 | 2 | 9 | 40 |
| Healthy lifestyles | 9 | 2 | 0 | 7 | 23 |
| Mental health | 5 | 4 | 0 | 1 | 15 |
| **TOTAL** | 40 | 15 | 5 | 20 | 116 |

**Table 3. Key mechanisms by study.**

| Authors & date | Peerness; similar, relatable and credible | A balance between autonomy and support | School values and broader change in school culture | Informal, innovative and personalised delivery methods | Friendship groups & closeness to a peer educator | Student nominated peers and peers as role models | Safe space – non-judgemental space to share experiences | Frequency of interaction and informal diffusion beyond the classroom | Ratio of peer educators to peer learners | Simplicity of health messages |
|---|---|---|---|---|---|---|---|---|---|---|
| **Alcohol, smoking, substance use** | | | | | | | | | | |
| Al-Sheyab, 2016 | | | ✓ | | | ✓ | | ✓ | | |
| Audrey, 2006 | ✓ | | | ✓ | ✓ | | | ✓ | | |
| Audrey, 2008 | ✓ | ✓ | | | | ✓ | ✓ | | | |
| Bloor, 1999 | | | | | ✓ | | | | | |
| Campbell, 2008 | | ✓ | | | ✓ | ✓ | | ✓ | ✓ | |
| Demirezen, 2019 | | | | ✓ | | | ✓ | | | |
| Dobbie, 2019 | ✓ | | ✓ | | ✓ | | | | | |
| LaChause, 2008 | | | | ✓ | | | | ✓ | | |
| Nurmala, 2020 | ✓ | ✓ | | | ✓ | | ✓ | | | |
| Perry, 1980 | | | ✓ | | | | | | | |
| Shah, 2001 | ✓ | | | | | ✓ | | | | |
| Starkey, 2009 | ✓ | | | ✓ | ✓ | ✓ | | | | |
| Weichold, 2012 | ✓ | ✓ | | | | ✓ | | | | |
| **Healthy lifestyles** | | | | | | | | | | |
| Ajuwon, 2019 | ✓ | ✓ | | | | | | ✓ | | |
| Bell, 2017 | | ✓ | ✓ | | | | | | | ✓ |
| Bogart, 2014 | | | ✓ | | | | | | | |
| Cui, 2012 | ✓ | | | | ✓ | | | | | |
| McQuinn, 2022 | ✓ | | | | ✓ | | | | | |
| Meyer, 2000 | | | ✓ | | | | | | | |
| Ping, 2014 | | ✓ | | ✓ | | ✓ | | | | |
| Sebire, 2019a | ✓ | ✓ | | | ✓ | ✓ | ✓ | ✓ | | |
| Sebire, 2019b | | ✓ | ✓ | | | | | | | |
| **Mental health** | | | | | | | | | | |
| Ellis, 2009 | | | ✓ | | ✓ | | ✓ | | | |
| Parikh, 2018 | | ✓ | | | | | | | | |
| Pickering, 2018 | | | ✓ | | ✓ | | | ✓ | ✓ | |

*(Continued)*

**Table 3.** (Continued)

| Authors & date | Peerness; similar, relatable and credible | A balance between autonomy and support | School values and broader change in school culture | Informal, innovative and personalised delivery methods | Friendship groups & closeness to a peer educator | Student nominated peers and peers as role models | Safe space – non-judgemental space to share experiences | Frequency of interaction and informal diffusion beyond the classroom | Ratio of peer educators to peer learners | Simplicity of health messages |
|---|---|---|---|---|---|---|---|---|---|---|
| Wright-Berryman, 2019 | | | ✓ | | ✓ | | ✓ | | | |
| Wyman, 2010 | ✓ | ✓ | ✓ | | | | | | ✓ | |
| **Sex education** | | | | | | | | | | |
| Caron, 2004 | | ✓ | | ✓ | | | | | | |
| Ito, 2022 | ✓ | ✓ | | | ✓ | | ✓ | | | |
| King, 2021 | ✓ | | | ✓ | | | ✓ | ✓ | | |
| Layzer, 2014 | ✓ | | | ✓ | | ✓ | | | | |
| Layzer, 2017 | ✓ | | | ✓ | | ✓ | | | | ✓ |
| Mason-Jones, 2011 | | ✓ | ✓ | | | | | | | |
| Mitchell, 2020 | ✓ | | | | | ✓ | | ✓ | ✓ | |
| Ozer, 1997 | ✓ | ✓ | ✓ | ✓ | | | | | | |
| Puentes, 2003 | | ✓ | | | | | | | | |
| Stephenson, 1998 | | | | ✓ | | | ✓ | | | |
| Strange, 2002 | | ✓ | ✓ | ✓ | | | | | | |
| Timol, 2016 | ✓ | ✓ | | | | | | | | ✓ |
| Visser, 2007 | | ✓ | ✓ | ✓ | | | ✓ | | | |

## Description of key mechanisms

**Peerness; similarity, relatability and credibility.** Nineteen studies discussed the similarity and relatability of the peer educators in relation to the success of peer education interventions, often defined as the concept of 'peerness'. [41, 44–46, 48, 52, 57, 59, 69–78].

Peers are viewed as sharing similar concerns and/or pressures, and often have similar experiences and insights related to health, and thus peers feel better able to communicate or mutually self-disclose health concerns to each other as opposed to a teacher or other adult.

Young people discussed feeling 'less awkward' getting information from someone closer to their own age when reflecting on a sex education intervention [46]. However, one study also recommended at least a two-year age gap between peer educators and peer learners due to trust issues with peers of the same age [74].

The extent to which the peer educators were demographically diverse, and similar to the peer learners they engaged with, was a key feature to ensure 'peerness', particularly with regards to age, gender, and lived experience of health condition/unhealthy behaviour. There was some disagreement across studies over whether peers should have experience of undesirable health behaviours. Specifically, the ASSIST Trial included peer educators who already smoked. The authors stated they could not be sure that regular smokers would effectively discourage non-smokers from smoking, but reflected that including smokers in the

**Table 4. Theories cited in school-based peer education health interventions.**

| Theories cited *(original citation)* | Included Studies Citing Theory |
|---|---|
| Social Learning Theory/Social Cognitive Theory [21, 22] | [42–51] |
| Diffusion of Innovation Theory [23] | [40, 41, 52–60] |
| The Health Belief Model [25, 61] | [41, 45, 46] |
| Principles of Positive Youth Development[62] | [45, 46] |
| Precaution Adoption Model [4] | [27] |
| Planned Behaviour [63] | [43] |
| Interpersonal Theory/Behaviour [64] | [43, 65] |
| Self-Determination Theory[66] | [40, 57] |
| Valente's Social Network Threshold Model[67] | [56, 58] |
| Empowerment Education [68] | [44] |

group of peer educators would help to engage 'smoking cliques' in the informal diffusion of knowledge [52].

Related to 'peerness' was the idea of peers being 'credible' sources of knowledge and young people being more likely to rely on peers for information. One study described young people seeing older peers as 'in the know' and more likely to be credible when providing advice and imparting knowledge [79]. Linking to the demographic diversity aspect, peer educators who are not representative of the diversity of peer learner groups were also seen to lack credibility.

**A balance between autonomy and support.**   Another common mechanism, also referred to in nineteen studies, related to the positive impact of peer educator autonomy, but a need to balance this with ongoing support from teachers throughout the intervention [40, 43, 48, 50, 53, 57, 69–71, 74, 75, 77, 80–85]. Support included both teacher presence within the session as well as peer educators being able to seek additional support outside of the sessions.

Autonomy referred to peers being given leeway in how they choose to carry out their role, but also to involving peer educators in designing the intervention. Greater autonomy for the peer educators typically meant adults ceding control, and consequently increased educator's feelings of investment, belief in the content and confidence in their abilities (see Weichold & Silbereisen, 2012). Peer educator autonomy also included peer educators identifying peers in need and using their own social channels to diffuse health information.

Several studies acknowledged that not all 'problems' or health discussions could be solved or managed by peer educators, and sometimes it was required to gain additional support form teachers to help find solutions or explanations [71]. One sexual health study found that although peer educators could share HIV information with their peers, they did not consider themselves competent to deal with issues such as rape and trauma which they felt required teacher involvement, additional training, and a reliable referral system. This finding was echoed by another paper where teachers suggested that peer education is likely to achieve better results for 'general health education knowledge'. But specialised health knowledge should be taught by teachers [83]. Another study discussed the potential problem of placing too much responsibility on the peer educators [74].

A further sexual health study described peer educators having difficulty gaining trust as a 'teacher' which took some time for their peers to get used to. Many peer educators described feeling anxious/unconfident at first which sometimes hindered their teaching abilities. [77].

*"When I taught one person, she said, 'Don't confuse me! Why do you need to teach me?"*

Studies also highlighted the importance of autonomy-supportive language to motivate behaviour change in peers by being encouraging and empathic without dictating what their friends should or should not do [40].

Some studies also discuss involving peer educators in designing or adapting content so it is more relevant to their peers [82].

Autonomy was also discussed in relation to the training of peer educators. In one physical activity study, the benefits of trainers providing choice, valuing peer educator input and using 'autonomy-supportive language' were discussed [57].

"*In (school) classes we just get a teacher telling us things, we were a lot more involved in what was going to happen and things like that.*" (*Peer Educator*)

The importance of peer educators being provided with adequate support in both training and delivery, was also discussed in relation to peers feeling overburdened, a potential negative consequence of peer education. Peer educators in one sex education study discussed the difficulty of having added responsibility, preparation and missing classes. This sometimes led to missing planning meetings and not being devoted to lesson planning [85].

**School values and broader change in school culture.**   Nine studies discussed the opportunity for peer education interventions to create broader system change within the school or changes in school culture [40, 42, 49, 51, 56, 58, 59, 65, 86] and six studies discussed the benefit of health messages being aligned with existing school values/norms [47, 48, 50, 55, 80, 84].

Studies discussed the culture of 'connectedness' and 'belonging' resulting from peer education interventions and these being carried forward through subsequent years of school, which in turn had a wider impact on school culture. Studies also discussed the positive impacts on broader shifts in school culture and building social cohesion across year groups of including more peer educators and peer learners, which sometimes led to higher 'reach' and increased health-related help-seeking amongst students. Some studies had a more specific focus on school-level changes, for example one suicide prevention intervention (Sources of Strength), focussed on changing norms across the full student populations through 3 months of school-wide messaging [58]. Additionally, this suicide prevention study found that trained peer leaders reported much more positive expectations that adults at school help suicidal students as well as increased norms for help-seeking from adults at school, perhaps therefore shifting help-seeking attitudes across the school more broadly.

Interventions were also found to be more effective if messages were consistent with existing school practice and policies. For example, one study described how in order to achieve effective promotion of health messages by peer educators, it is necessary that the intervention fits with the views of the teachers and the existing school culture, to promote the agency of staff and students in pursuing better health [84]. This also included the involvement of teachers in the programme as well as the existence of school-based policy that supported the intervention messages.

One evaluation of a smoking intervention also suggested that peer interventions can counter existing peer culture within a school, particularly given that the adoption of smoking habit is inherently social [49].

**Informal, innovative and personalised delivery methods.**   Thirteen studies discussed the importance of peer educators delivering the intervention in a personalised, informal and dynamic style [27, 43, 45, 46, 48, 50, 52, 60, 78, 83, 84, 87, 88]. Linking to the importance of similarity of experiences and views in the 'peerness' mechanism, studies discussed the benefit of peers exchanging personal stories and experiences, which were thought to be more salient for the peer learners [48]. Equally, this mechanism relates to peer educators being given the autonomy and freedom to be able to deliver lessons in this manner.

Peer interventions often involve multiple learning modalities, which were often more creative and engaging than traditionally structured teacher-student lessons. For example, in an intervention that focused on using short comedy sketches to convey sexual health messages [46], the peer educators discussed how the least successful workshop was the '*least interactive most lecture-like – like feeding them information*'. Similarly, a paper on sun safety discussed learning through interesting and interactive activities being more memorable [83]. One study also described how the novel content of the lessons led curiosity among peer learners [88]. Sharing personal stories and experiences is also a unique learning modality in comparison to typical lessons.

One study evaluating the prevention of foetal alcohol syndrome reflected on the lack of personal and dynamic delivery used which lead to weakened effects of their intervention due to its 'highly didactic instructional approach' and lack of interactive experiences to teach adolescents the skills they need to avoid unhealthy risk behaviours [27]. The authors of this study reflected that this particular intervention is largely a biomedical presentation and lacks the interactive experiences shown to be effective when teaching adolescents to avoid risky behaviours.

Studies discussed a general preference for the informal delivery of peer educator sessions which led to students feeling more relaxed than they would with teachers and being able to be more open. One study discussed how they felt a shift from 'talking at' to 'chatting to'. However, this sometimes led to a tension between maintaining informality and exerting authority/control over the class [50]. Students often identified that successful activities tended to be practical, involving moving around and having 'fun'.

The removal of the typical teacher-student power dynamic/authoritarian relationship was also discussed in terms of how peers are seen as social equals. When asked how peer educators interacted with peer learners, in a sexual health intervention study, one peer educator answered: "*. . .just how we usually speak. . .we didn't speak to them like year 9s, we spoke to them like sort of equals*".

One paper discussed how when peers used more disciplinary approaches (for example asking why students were not engaging in the session), this led to disengagement from peer learners [48], perhaps as this reflected a more typical teacher-student relationship. Similarly in a sex education intervention, peer educators noted that the more they channelled the role of a traditional health teacher, the less effective they felt they were as workshop facilitators [46]

**Friendship groups and closeness to a peer educator.**    Thirteen studies discussed the role of friendship and closeness between peer educators and peer learners [44, 52–54, 56, 57, 59, 60, 65, 76, 77, 86, 89]. One study (mental health; mindfulness) discussed how familiarity with their peers could boost motivation to engage with the intervention, feel more relaxed, and therefore learn more quickly [90].

A sexual health study described how friendships between peer educators and peer learners led to increased motivation, and how close relationships can aid in teaching and learning [77].

"*When they first came to teach us, we felt good because they are our friends, they are always with us, they listen to us, and they understand us. Therefore, we understood what they were teaching us, and we tried to follow what they taught us.*"

A smoking intervention discussed how peer education interventions are likely to be best introduced into close-knit groups characterised by intra-group interaction, endurance of peer-peer relationships, and likelihood for peers to stay in touch once the intervention has ended, which in turn enhanced the sustainability of the intervention effects [44, 53]. Other studies discussed being motivated to engage in activities through familiarity and friendship and how

receiving support from close friends was more likely to be accepted and met with a positive attitude [57].

One study (mental health; suicide prevention) also demonstrated the importance of personal affiliations to peer leaders and natural friendship networks as a medium for promoting peer-led prevention efforts, finding that having a friend who was a peer leader led to higher rates of intervention exposure [56].

As well as making use of existing social networks, interventions also provided opportunity for engaging with peers outside friendship cliques in order to spread health messages further. One evaluation of a smoking intervention demonstrated the opportunity for peers to make new friends as a result of the intervention, describing getting to know people they 'wouldn't usually mix with', and offering the opportunity for peers to relay health messages to the wider school group [59].

One physical activity study also discussed peer learners being more open with peer educators if it were a conversation between friends [57].

"*When it's coming from a friend or someone that they're close with then they're more sort of open about being active and what they would like to do, maybe rather than . . . So I feel that it's sort of better coming from a friend.*"

Another study suggested that pairing environmental changes with education and awareness raising among adolescents is more likely to lead to a change in behaviour [55].

Many intervention descriptions also discussed the peer education intervention influencing students at both the classroom and school level [47].

**Student nominated peers and peers as role models.**   Peers acting as role models, and the importance of selecting peer educators looked up to by other students was discussed in 11 studies [41, 42, 45, 46, 53, 57, 60, 70, 72, 75, 83].

"*Peer educators are nice people who we can look up to, so the material sank in better*" [46].

Teachers from a study evaluating a sun safety intervention discussed how peer education can promote students to set an example for others, and then peer learners are more likely to accept it as normal/desirable [83].

Peer educators seen as role models by their peers can model 'appropriate' health behaviours. Staff reflected on the importance of peer nomination (also discussed as a separate mechanism) in terms of selecting role models for peers. One study discussed how peers nominated '*influential individuals who have an effect on other people, and other people look up to and see as leaders, or people they aspire to*' [60].

Many studies adopted a peer-nomination approach, whilst others relied on staff to nominate students. One study that used staff nomination asked staff members to nominate up to six students whose '*voices are heard*' by others students [56]. The prestige of peer nomination also led to peer educators being viewed as more credible [53].

Some studies also discussed staff and students views of the peer nomination processes. Teachers in some studies acknowledged that the peer nomination approach resulted in a diverse group of students (which has been acknowledged as important within the 'peerness' mechanism) some of whom were unlikely to have been selected by school staff [60]. Students sometimes expressed reservations, often due to some peer educators lacking confidence or not taking the sessions seriously. As well as role modelling appropriate health behaviours, student nominated peers also led to some concerns regarding peer educators role modelling inappropriate

health behaviours. For example, in a smoking intervention study, peer educators who smoked were thought to be an asset by some students but viewed as hypocritical by others [60].

**Safe and non-judgemental space to share experiences.**   Ten studies discussed peers and/or external trainers creating a safe space to share their thoughts and feelings relating to health [57, 65, 70, 71, 77, 78, 84, 86–88]. Students described having the opportunity to share information that they would be uncomfortable sharing with adults, without fear of judgement.

Students in one sexual health intervention study described teachers being reserved and less explicit in comparison to the openness of peer educators [50].

*"I think teachers are quite reserved in what they would say and how explicit they would go. They [Peer Educators] are not really worried about what they say to us...it was quite open."*

Non-judgemental spaces also related to the facilitators who trained the peer educators, for example teachers in one study reflected on how students were likely to feel judged if they raised a point about them or their peers smoking if a teacher led the training, but this was reduced by having an external organisation involved with health promotion trainers and youth workers and teachers asked to take a more passive role [70]. External trainers in another study also discussed how peer educators felt they could trust them [57].

"*I think it was nice that they could probably talk to us and they know that we, you know, they could trust us. We wouldn't go in and talk about them behind their back and I think, and I could trust them as well.*" (Trainer)

However, one sexual health intervention found that as sexual health matters were not talked about openly in schools, peer learners were initially very shy to share their opinions. This particular study had to run separate sessions for boys and girls to facilitate discussion[45].

Peer learners in another sexual health intervention discussed the ability to talk openly on sensitive issues which they felt unable to do in typical school lessons with teachers [88].

"*You could speak like normally like you would with your friends about stuff, you weren't frightened that you'd use a bad word.*" Another said "*I don't think you get the chance to talk a lot in other classes. I don't know, it's more difficult to just speak out on like sensitive things.*"

Providing a safe, non-judgemental ear for listening was also discussed in the Hope Squad suicide prevention program on the basis that "*kids tell kids*" when they are suicidal and peer support could help ameliorate social disconnectedness [65].

**Frequency of interaction and informal diffusion beyond the classroom.**   Nine studies discussed the frequency of interaction between peer educators and peer learners [27, 41, 42, 52, 53, 56, 57, 69, 78]. This included the benefit of multiple sessions and repetitive delivery of health messages but also the opportunity peer education provides to continue relaying positive health messages beyond the classroom. Studies discussed how peers can continue the conversation with each other outside of lessons, which demonstrated an advantage of peer diffusion of health-related knowledge in comparison to a traditional teacher-student classroom context which is constrained to one single lesson.

Peer educators in one sexual health study discussed being available after the lesson to support students with specific needs, for example specific questions about sexual health concerns or how to access contraception. Peer educators therefore provided direct health-related help for peer learners outside of lessons [78].

"*Like a lot of them [participants] are wanting condoms, and so [The Health Program] pro-vided the condoms. Even when we ran out of condoms and they asked for some, we referred them to the closest health department in our area.*"

This mechanism is also related to the peer educator autonomy mechanism as well as the mechanism of friendship groups and closeness to peer educators. Particularly for interventions that rely on information diffusion outside of the classroom as this involves peer educators act-ing autonomously in terms of both who they target, what information they impart and how as well as existing relationships outside of the classroom making interactions more likely.

Some peer learners in a physical activity intervention, discussed not feeling they had received any 'support' from peers and peer educators believed this may be due to the informal ways they gave support; "*They didn't really know I was peer supporting because with some friends we did it quite subtly.*" (Peer Educator) [57].

**Ratio of peer educators to peer learners.** Four studies discussed the importance of the number of peer leaders being trained, particularly in interventions relying on information dif-fusion across the school [41, 53, 56, 58]. One such study focussed on suicide prevention and found training more peer leaders increased school-wide exposure [56]. This study also found that training up to 15% of the student population as peer leaders increased intervention expo-sure but after this, the effect appeared to level off.

Similarly, a smoking prevention trial (ASSIST), also discussed training 15% of the target group to maintain a so-called 'critical mass' of peer educators. This proportion had been discussed in previous literature around HIV prevention. This critical mass was also dis-cussed in a sexual health study involving informal diffusion. Given the sensitive topic, they increased the proportion of peer nominated students to 25% of the year group during recruitment [41].

Another suicide prevention intervention (Sources of Strength) discussed how the ratio of peer educators to peer learners should be addressed in future studies as the optimal proportion of students to train as peer educators remains unanswered. This was particularly discussed in reference to the impact the ratio would have in smaller schools which may have different social norms which could dissuade disclosure of suicidal behaviour (i.e. if there were too many peer educators encouraging peers to seek support from adults this may dissuade help-seeking behaviours altogether) [58]. As a core focus of the Sources of Strength intervention is to engage 'trusted adults' to help distressed and suicidal peers, the balance of peer educators to peer learners as well as the mechanism of balance between peer educator autonomy and staff sup-port are likely to be particularly relevant to this particular intervention

**Simplicity of health messages.** Two studies reflected on the need for health messages to be simple, particularly for interventions relying on informal diffusion of information [74, 80]. One study modelled on informal diffusion of knowledge had a dual focus of physical activity and healthy eating, however the study deemed the dual focus '*too complex for information dif-fusion through adolescent peer networks*'. The study concluded that there is a need for health messages to be simple for trainers to teach and students to pass on. However, a tension was also highlighted between a desire not to oversimplify or isolate health behaviours and the need to present clear succinct health promotion messages.

In contrast to this, students in one study evaluating a sexual health intervention discussed the need for depth in health promotion messages. One young person reflecting on sex educa-tion in schools and criticising the lack of depth in lessons delivered by teachers with the main emphasis of the lesson being "don't do it, and that was basically it" [46]. In a control school of another study evaluating a sexual health intervention, feedback from students in a control school also criticised teachers for being repetitive and repeating the same content. "*We do the*

*same subject every time, all about puberty and development. I don't think they can think of anything else to teach us.*" [88].

Given this contrast, it may be that there are differing views between teaching staff and students on how complex and in-depth health messaging should be. It appears that students feel that more in-depth discussion is more likely to lead to a change in behaviour in comparison to simple and repetitive health messaging.

## Mechanisms by health area

We identified the most common mechanisms per health area. For alcohol, smoking and substance use, 'peerness' (n = 7) was the most cited. For healthy lifestyles, peerness (n = 4), peer educator autonomy vs. additional support from teachers (n = 4) and school values/broader change in school culture (n = 4) were equally cited. For sex education, a balance between peer educator autonomy and teacher support (n = 8), and information/personalised delivery methods (n = 8) were the two most cited mechanisms, followed by 'peerness' (n = 7). 'Broader system/cultural change' (n = 4) was the most cited mechanism for mental health interventions followed by friendship groups/closeness to a peer educator (n = 3).

## Discussion

This review aimed to identify the key mechanisms in health-based peer education interventions in school settings. We identified 10 key mechanisms from the literature across four health areas. Health areas covered in this review demonstrate that mechanisms of school-based peer led interventions have been explored predominantly within alcohol, smoking and substance use, healthy lifestyle interventions and sexual health interventions, which follows the pattern of our effectiveness review [38].

With regard to providing support for peer educators, one study discussed the importance of close adult supervision, but also the need for resources and a reliable help-seeking/referral pathway to be in place to support the intervention so peer educators do not become overburdened or feel they are letting their peers down [84]. Close adult supervision and established help-seeking pathways are likely to be important requirements for future peer education interventions.

Several studies discussed the need for peer educators to be sufficiently similar to their classmates to enhance credibility, however as identified in existing studies [81], future research needs to take this further in developing more detailed definitions of 'peerness' to understand in what ways peer educators need to be similar to peer learners for example defining specific characteristics which could in turn refine and improve the selection method of peer educators to give interventions maximum impact. Furthermore, the ideal age gap between peer educators and peer learners requires further research. Studies differed in how similar in age peers should be and it is possible that some degree of social distinction (e.g. age) is important for credibility.

The role of friendship groups and closeness to peer education mechanism is consistent with previous findings from peer-led programs and theoretical models [56, 91] as well as the need for informal and interactive experience as a means to teach adolescents skills they need to avoid unhealthy risk behaviours [92]. It is likely that there are important links between some of the key mechanisms identified, for example if peer educators are granted autonomy, this may naturally lead to a more informal, interactive and personalised delivery of lessons and avoid didactic approaches that were found to be unhelpful [27].

While multiple studies reported suggested mechanisms which led to improved health literacy and/or behaviour, many failed to demonstrate this explicitly in their findings and often authors reflected on quantitative findings using existing theoretical assumptions. This gap

requires more studies to carry out robust process evaluations or to use realist approaches that explicitly focus on mechanisms of change so we can continue to understand better the pathways to impact on different health outcomes.

A key implication of the paper is the need for logic models and theories of change which underpin peer education interventions for health. We found a tendency for papers to discuss peer education mechanisms more generically, rather than as they relate to specific health outcomes. For example, a number of key mechanisms were discussed in relation to why peer education works as an approach, without linking this to why it works for health outcomes. Further research is required to understand how applicable and important these more generic peer education mechanisms may be when applied to specific health interventions or outcomes. Creating detailed logic models would help deal with this missing link to more clearly map out how specific intervention components led to particular health outcomes. The importance of logic models and theories of change have been highlighted within the new Medical Research Council Guidance regarding health interventions and the importance of detailing how interventions are expected to work. [93] It is likely that different mechanisms are more or less important for different health outcomes, leading to different peer education approaches (e.g. informal diffusion of knowledge versus formal classroom/taught approaches).

Another interesting finding was that at least one mechanism (simple health messages) was found to make implementation of the intervention easier, but perhaps reduced intervention effectiveness. This example again highlights the need for future studies to more clearly theorise and test mechanisms that are being activated and how these relate to effectiveness. One opportunity realist evaluation provides is to identify potential problems or components of interventions that may be unhelpful. One potential harm raised in this review was the pressure and responsibility placed on peer educators, which was particularly highlighted through the mechanism of finding a balance between peer educator autonomy and teacher support. Providing peer educators with full control may risk inappropriate or potentially harmful content being passed on to peer learners, but it also risks peer educators feeling burdened by peer disclosures or feeling left not able to talk about their own health concerns. Pressure on peer educators has been discussed in the wider literature with regard to a number of issues including peer educators having to deal with personal questions about their own experiences, potential hostility from members of their peer group, reduced confidence when unable to manage difficult situations, as well as frustration when peer expectations are not met and feeling unable to address their own problems or seek help [94–97]. A further implication of this review therefore will be unpicking potential harms of peer education and minimising these harms through well thought out logic models, for example how can we minimise pressure on peer educators and fully support them to teach their peers, whilst allowing them a level of autonomy and freedom to maintain benefits observed from these mechanisms.

One study included in this review concluded that peer education 'would seem to be a method in search of theory, rather than the application of theory to practice' [83]. Despite a number of papers being identified that evaluated peer education interventions to improve young people's health [38], relatively few of these explicitly focused on mechanisms relating to intervention effectiveness. There is a clear need for future research and intervention evaluations to focus on change mechanisms for example through the development of logic models that link program activities to anticipated results; a priority mentioned in one included study within this review [27]. Logic models will in turn provide an increased focus on pathways to outcomes to better understand how and why peer education interventions can improve adolescent health outcomes. It is still unclear whether there are distinct mechanisms at work in each health area, or if generic peer education mechanisms will apply across all health areas.

## Limitations

It is likely that some relevant studies may have not been picked up by our search terms due to the wide range of definitions of 'peer education' and multiple variations of this term. We did not search grey literature as part of this review which is another possible area that may have picked up additional relevant studies. This review only focused on universal peer education which targeted whole classes or year groups, it may be the case that there is more literature on key mechanisms involved in more targeted peer education interventions that focus on specific at-risk groups. Studies that did not meet our inclusion criteria (e.g. wider age ranges, out-of-school settings), may have included information about mechanisms that have not been identified within this review.

A number of studies excluded from this current review focused on the impacts of participating in peer education on peer educators themselves. These papers were excluded as they were focussed on non-health outcomes such as increased empathy, confidence and self-esteem in peer educators, rather than why peer education interventions are effective for improving health. Another implication of this review is the need for a future focus on the impact of being a peer educator on a number of social outcomes, perhaps taking a similar realist approach to better understand this impact, which may also include any risk of harm and the testing of dark logic models [98].

## Conclusion

This review identified 10 key mechanisms of peer education interventions across four key health areas (alcohol, smoking and substance use, sex education, healthy lifestyles and mental health). This review is the first realist-informed study to synthesize key mechanisms of school-based peer education seeking to improve health literacy and health behaviours across all health areas.

To further our understanding of the active components of school-based peer education interventions for health improvement, more process or realist evaluations that focus on mechanisms of change are required. Although several mechanisms were identified in this review, many were drawn from theoretical assumptions that were never tested or evaluated, and no papers were found to hypothesise mechanisms through logic models or context-mechanism-outcome configurations and then test these mechanisms within their evaluations. Future studies of peer education interventions should focus not only on the mechanisms at play, but how these relate to specific health outcomes, as well as the contextual factors that may constrain or help the mechanisms to be activated, and which ultimately impact on the effectiveness of an intervention.

## Pre-registration

This review was pre-registered on PROSPERO: CRD42021229192 (accessible here: https://www.crd.york.ac.uk/prospero/display_record.php?RecordID=229192). One deviation was made from the original protocol which was the use of a different quality appraisal tool. Initially we had planned to use the Canadian Effective Public Health Project Practice (EPHPP) Quality Assessment Tool for Quantitative Studies and the Critical Appraisals Skills Programme (CASP) checklist for qualitative studies. The authors instead used a combined mixed methods tool (the Mixed Methods Appraisal Tool; MMAT) for both quantitative and qualitative studies. This was due to the large volume and variation of studies which meant there were benefits to using a single brief quality check tool across all included studies, allowing us to standardise scores across study types.

## Supporting information

**S1 Table. Prisma checklist.**
(DOCX)

**S2 Table. Full search strategy.**
(DOCX)

**S3 Table. Overview of included studies.**
(DOCX)

## Author Contributions

**Conceptualization:** Emily Widnall, Mark Limmer, Judi Kidger.

**Data curation:** Emily Widnall, Steven Dodd.

**Formal analysis:** Emily Widnall, Steven Dodd.

**Investigation:** Emily Widnall, Steven Dodd.

**Methodology:** Emily Widnall, Steven Dodd.

**Project administration:** Emily Widnall, Steven Dodd.

**Resources:** Emily Widnall.

**Software:** Emily Widnall.

**Supervision:** Emily Widnall, Judi Kidger.

**Validation:** Emily Widnall.

**Visualization:** Emily Widnall.

**Writing – original draft:** Emily Widnall.

**Writing – review & editing:** Emily Widnall, Steven Dodd, Abigail Emma Russell, Esther Curtin, Ruth Simmonds, Mark Limmer, Judi Kidger.

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
