## [Decision Letter · Decision Letter 0]

28 Sep 2023

PONE-D-23-18256Mechanisms of school-based peer education interventions to improve young people’s health: a realist-informed systematic review.PLOS ONE

Dear Dr. Widnall,

Thank you for submitting your manuscript to PLOS ONE. After careful consideration, we feel that it has merit but does not fully meet PLOS ONE’s publication criteria as it currently stands. Therefore, we invite you to submit a revised version of the manuscript that addresses the points raised during the review process.

Check comments from the two reviewers and respond to them. 

A marked-up copy of your manuscript that highlights changes made to the original version. You should upload this as a separate file labeled 'Revised Manuscript with Track Changes'.An unmarked version of your revised paper without tracked changes. You should upload this as a separate file labeled 'Manuscript'.

We look forward to receiving your revised manuscript.

Kind regards,

Sogo France Matlala, PhD

Academic Editor

PLOS ONE

https://journals.sagepub.com/doi/10.1177/001789690306200107

https://www.tandfonline.com/doi/full/10.1080/17290376.2016.1241188

https://www.ncbi.nlm.nih.gov/pmc/articles/PMC1758164/pdf/v074p00405.pdf

https://www.tandfonline.com/doi/abs/10.1080/15546128.2021.1959472?journalCode=wajs20

In your revision ensure you cite all your sources (including your own works), and quote or rephrase any duplicated text outside the methods section. Further consideration is dependent on these concerns being addressed.

 “School for Public Health Research

(PD-SPH-2015)”

Reviewers' comments:

Reviewer's Responses to Questions

**Comments to the Author**

1. Is the manuscript technically sound, and do the data support the conclusions?

Reviewer #1: Yes

Reviewer #2: Partly

2. Has the statistical analysis been performed appropriately and rigorously? 

Reviewer #1: N/A

Reviewer #2: N/A

3. Have the authors made all data underlying the findings in their manuscript fully available?

Reviewer #1: No

Reviewer #2: Yes

4. Is the manuscript presented in an intelligible fashion and written in standard English?

Reviewer #1: No

Reviewer #2: Yes

5. Review Comments to the Author

Reviewer #1: he authors should be commended for the interesting topic on systematic review. I have some suggestions to improve it:

1. Introduction:

o Well-written novelty was clearly outlined.

o However, this is a literature review study and the authors have outlined many studies at the end, they cited one author e.g. Many assumptions on the effectiveness of peer education for health improvement centre around 74 adolescents. For example, discussing adolescence as a time for consolidating health-related values,75 attitudes and lifestyles and making decisions about a variety of behaviors which have important 76 consequences for future health [8], as well as discussion of an increase in risk-taking behavior in 77 adolescents [9]. Many also argue that peer education is based on the rationale that peers have a 78 stronger influence on individual behavior due to the level of familiarity and trust and the comfort 79 they are able to provide [10].

o Different referencing styles were used e.g (Greenhalgh et al., 2007, Kaneko, 1999, O’Campo et al., 2009) and [10].The authors must align their referencing with the ones recommended by the journal.

2. Methods

o Inclusion criteria: Authors should indicate the rationale of only including 2020 to 2023 studies.

o Furthermore, this section is not clear some studies in the reference list dated back to 1997.

3. Results:

o Well presented

o However authors indicated that studies also highlighted the importance of autonomy-supportive language to motivate behavior change in peers by being encouraging and empathic without dictating what their friends should or should not do [34]. But one author was cited in which studies??

o

4. Discussion: Well written

o However, authors need to discuss the implications of the findings and also create arguments based interventions of peer interventions on health outcomes might differ based on several factors context, countries , socioeconomic status, resources, etc

5. Conclusion:

o This section should summarize the important findings and outline the novelity of the study, what new (What is the novelty of the study and its scientific value?)) and what are the recommendations

6 .I have reviewed this paper from a scientific/technical perspective, and while doing so have also reviewed the language and syntax. The paper is well written.

Reviewer #2: I believe this to be an important topic to explore given the use of peer education across school and community settings. It has been interesting to get the opportunity to review this article.

I think, however there is considerable room to improve this article, please see my suggestions below. Some parts of the manuscript had line numbers which I have used to reference my comments, others didn’t, so I have used page numbers when referring to latter parts of document.

General feedback:

1. I believe the word count could be considerably reduced through greater conciseness. An example of this is within rows 113-124 and 101-103 which say very similar things. Another example is found between rows 167 to 171. Furthermore, rows 219-223 are possibly unnecessary to include if including table, just signpost readers to table.

2. Punctuation and spelling should also be double checked throughout document. Two examples are in row 221 sentence ends midway and row 232 where Bandura is spelled incorrectly.

3. I would recommend checking both the literature used for background and included in the studies to ensure that references to points made or quotations are linked specifically to the mechanisms involved in “school based peer education” rather than that of the peer educators being trained. For example, this is unclear in rows 77-79 and with use of reference #51.

4. The use of the referencing system changes e.g. row 122. Also in row 122 the article discusses recent reviews but the references provided are from 1999 – 2009. Check that all references are in line with PLOSone requirements.

5. This article has discussed mechanisms of change however has not linked these to outcomes of the studies. Row 150-151 suggested that studies were to include appraisal of mechanism and outcome but this was not reported upon. It would be important within the article for the reader to gain understanding about the mechanisms at work, but also how this links to the success or failure of the intervention's impact on health literacy or health behaviour outcomes relating this to the study context. For example, when peer educators reported being given too much responsibility or felt unable to field questions presented by peer learners did this in turn correlate with less impact on health literacy or adoption of desired healthy behaviours. Where there factors at play in these settings which exposed the peer educators more rather than other settings where this was not reported or reported as not an issue?

Title:

6. Further clarity could be gained by including “health literacy and health behaviours” in the title. Suggestion: “Mechanisms of school-based peer education interventions to improve young people’s health literacy or health behaviours: a realist-informed systematic review.”

Abstract:

7. Identify health literacy and health behaviour concepts. This clarity could be added at points throughout document.

Background:

8. It would be helpful to read about the importance of the topic… why is this a topic worthy of research? This will help set the scene and point readers to the usefulness of what is to follow.

Methods:

9. Row 148 mentions that studies were either qualitative or quantitative but the results section suggests there were 20 mixed methods studies.

10. Inclusion /exclusion: It might be helpful to consider use of a table to present the inclusion / exclusion criteria.

a. In limitations, the article mentions that only studies pertaining to a “whole class or year group” were included, however this is not mentioned in the inclusion criteria.

b. Earlier the article refers to including studies that focus on health literacy and health behaviours. Point 3 (row 145) refers to health-related outcomes (health knowledge, attitudes, or behaviours). Consistency of concepts will help reader and focus study.

c. Row 161, should also exclude counselling intervention as this was mentioned earlier as beyond scope of study.

11. Rows 167 to 171 … Where their discrepancies between reviewers? What were these? How were they resolved?

12. What type of thematic analysis was used? What was the process of theme development?

13. Rows 192-198 I don’t believe are necessary. For transparency the MMAT could be attached as supplementary material displaying the criteria which was used to inform decision making.

14. Should sentence in row 180 be combined with sentences in rows 172-174.

Results:

15. Referencing style changes in tables.

16. Figure 1 may be better located in methods section.

17. Rows 215-218 maybe belong better in methods section.

18. HIV/AIDS prevention. It would be helpful to not have these as interchangeable acronyms. HIV prevention would be suffice.

19. Pg 11. Did studies with theories differ in quality of mechanisms used to others without?

20. I have not provided extensive feedback on the themes. I see similarities across some of the themes and believe that there needs to be further analysis conducted which could considerably reduce the number of themes but also enhance the story by linking understanding of the mechanisms of change to outcomes in studies. I think the analysis requires further work and this will allow this article to provide the reader with richer understanding of the phenomenon. I see potential links in themes 1, 5 and 6 and again in 4, 8 and 10. There is repeated information through themes, however I believe further analysis and structure applied could really help reduce this and provide a clearer picture.

Discussion

21. Given my comments on the revised thematic analysis I have no comments on the discussion at this time. I look forward to reading how the findings tie up with the objectives and how this informs the reader about potential mechanisms of change within school based peer education programmes that can promote health literacy and adoption of health behaviours.

6. PLOS authors have the option to publish the peer review history of their article (what does this mean?). If published, this will include your full peer review and any attached files.

Reviewer #1: No

Reviewer #2: No

---

## [Author Response · Author response to Decision Letter 0]

28 Mar 2024

Reviewer #1: The authors should be commended for the interesting topic on systematic review. I have some suggestions to improve it:

1. Introduction:

Well-written novelty was clearly outlined. However, this is a literature review study and the authors have outlined many studies at the end, they cited one author e.g. Many assumptions on the effectiveness of peer education for health improvement centre around 74 adolescents. For example, discussing adolescence as a time for consolidating health-related values,75 attitudes and lifestyles and making decisions about a variety of behaviors which have important 76 consequences for future health [8], as well as discussion of an increase in risk-taking behavior in 77 adolescents [9]. Many also argue that peer education is based on the rationale that peers have a 78 stronger influence on individual behavior due to the level of familiarity and trust and the comfort 79 they are able to provide [10].

The authors thank the reviewer for noting this. We confirm that multiple studies/authors have been cited which speak to different individual points for example one study relating to consolidating health values/attitudes and lifestyles, another study regarding risk-taking behaviour and another study relating to familiarity and trust.

Different referencing styles were used e.g (Greenhalgh et al., 2007, Kaneko, 1999, O’Campo et al., 2009) and [10].The authors must align their referencing with the ones recommended by the journal.

The authors thank the reviewer for highlighting this inconsistency, the references have now been revised to match the rest of the manuscript.

2. Methods: Inclusion criteria: Authors should indicate the rationale of only including 2020 to 2023 studies. Furthermore, this section is not clear some studies in the reference list dated back to 1997.

The authors confirm that there was no limit on publication dates. Original searches took place in 2020 but authors additionally conducted a second set of searches between 2020 to 2023 to ensure the review was up to date with current literature, this has now been clarified in the abstract. 

3. Results: Well presented, however authors indicated that studies also highlighted the importance of autonomy-supportive language to motivate behavior change in peers by being encouraging and empathic without dictating what their friends should or should not do [34]. But one author was cited in which studies?? 

The authors thanks the reviewer for highlighting this discrepancy. The language has now been changed to reflect that one article noted the important of autonomy-supportive language rather than multiple studies.

4. Discussion: Well written. However, authors need to discuss the implications of the findings and also create arguments based interventions of peer interventions on health outcomes might differ based on several factors context, countries , socioeconomic status, resources, etc

The authors thank the reviewer for this comment. We have revised the discussion to refine focus based on implications of the findings. The three key implications highlighted are:

1. A key implication of the paper is the need for logic models and theories of change which underpin peer education interventions for health. We found a tendency for papers to discuss peer education mechanisms more generically, rather than as they relate to specific health outcomes.

2. A further implication of this review therefore will be unpicking potential harms of peer education and minimising these harms through well thought out logic models, for example how can we minimise pressure on peer educators and fully support them to teach their peers, whilst allowing them a level of autonomy and freedom to maintain benefits observed from these mechanisms.

3. Another implication of this review is the need for a future focus on the impact of being a peer educator on a number of social outcomes, perhaps taking a similar realist approach to better understand this impact, which may also include any risk of harm and the testing of dark logic models [98].

In terms of context factors, given that this is a novel realist review of mechanisms, the authors do not feel in a position to discuss how mechanisms relate to context, countries and socio-economic status, but we have now included how these factors will be important for future research to explore more fully. We also acknowledge that effective mechanisms of peer education are likely to be context and health area specific. 

5. Conclusion

This section should summarize the important findings and outline the novelty of the study, what new (What is the novelty of the study and its scientific value?)) and what are the recommendations.

The authors thank the reviewer for suggestions on refining the conclusion. We have now re-written the conclusion to summarize key findings and address the novelty of the study.

6. I have reviewed this paper from a scientific/technical perspective, and while doing so have also reviewed the language and syntax. The paper is well written.

The authors thank the reviewer for this comment.

Reviewer #2

I believe this to be an important topic to explore given the use of peer education across school and community settings. It has been interesting to get the opportunity to review this article.

I think, however there is considerable room to improve this article, please see my suggestions below. Some parts of the manuscript had line numbers which I have used to reference my comments, others didn’t, so I have used page numbers when referring to latter parts of document.

The authors thank the reviewer for their positive comments on the manuscript. We have addressed each of your suggestions point-by-point below.

General feedback:

1. I believe the word count could be considerably reduced through greater conciseness. An example of this is within rows 113-124 and 101-103 which say very similar things. Another example is found between rows 167 to 171. Furthermore, rows 219-223 are possibly unnecessary to include if including table, just signpost readers to table.

The authors thank the reviewer for this comment. We have now revised the noted examples to be more concise.

2. Punctuation and spelling should also be double checked throughout document. Two examples are in row 221 sentence ends midway and row 232 where Bandura is spelled incorrectly.

The authors have now completed a thorough proofread of the final manuscript and punctuation and spelling errors have been corrected where necessary.

3. I would recommend checking both the literature used for background and included in the studies to ensure that references to points made or quotations are linked specifically to the mechanisms involved in “school based peer education” rather than that of the peer educators being trained. For example, this is unclear in rows 77-79 and with use of reference #51.

The authors thank the reviewer for this comment. The paragraph mentioned related to the period of adolescence more broadly and more generic rationales for peer education to add context to the review. The reference highlighted discusses mechanisms relating to peer nomination and information diffusion.

4. The use of the referencing system changes e.g. row 122. Also in row 122 the article discusses recent reviews but the references provided are from 1999 – 2009. Check that all references are in line with PLOSone requirements.

The authors have now refined the references to all be consistent and meet PLOSone requirements. The authors also thank the reviewer for noting the older reference used which has now been revised to include a more recent example of a realist review for mental health interventions.

5. This article has discussed mechanisms of change however has not linked these to outcomes of the studies. Row 150-151 suggested that studies were to include appraisal of mechanism and outcome but this was not reported upon. It would be important within the article for the reader to gain understanding about the mechanisms at work, but also how this links to the success or failure of the intervention's impact on health literacy or health behaviour outcomes relating this to the study context. For example, when peer educators reported being given too much responsibility or felt unable to field questions presented by peer learners did this in turn correlate with less impact on health literacy or adoption of desired healthy behaviours. Where there factors at play in these settings which exposed the peer educators more rather than other settings where this was not reported or reported as not an issue?

The authors thank the reviewer for highlighting this. Although the aim of this review was to identify key mechanisms of change in school-based peer education, we found that studies included in the review tended to talk about mechanisms in general, rather than linking these mechanisms to specific outcomes. We make this point in the discussion, just before the limitations section. One key finding for example was the lack of logic models or programme theory (only found in two papers) which clearly illustrate how mechanisms lead to a change in outcome. This meant that we were unable to say more in this paper about specific mechanisms linking to specific outcomes, and also specific contextual factors that may influence how effective an intervention may be. We do agree this is an important next step for future research in this area, and we have added this point more clearly in our final conclusion. 

We focused on effectiveness and outcomes specifically in our separate effectiveness review paper: https://bmcpublichealth.biomedcentral.com/articles/10.1186/s12889-022-14688-3

Additionally, we provided reflection on how one mechanism was thought to make implementation easier, but may reduce the effectiveness of the intervention:

Another interesting finding was that at least one mechanism (simplicity of health messages) was found to make implementation of the intervention easier, but perhaps reduced intervention effectiveness. This example again highlights the need for future studies to more clearly theorise and test mechanisms that are being activated and how these relate to effectiveness.

Title:

6. Further clarity could be gained by including “health literacy and health behaviours” in the title. Suggestion: “Mechanisms of school-based peer education interventions to improve young people’s health literacy or health behaviours: a realist-informed systematic review.”

The authors thank the reviewer for this comment, we have changed the title accordingly.

Abstract:

7. Identify health literacy and health behaviour concepts. This clarity could be added at points throughout document.

The authors thank the reviewer for this comment regarding clarity, we have now added additional mention of health literacy and health behaviour concepts in the abstract.

Background:

8. It would be helpful to read about the importance of the topic… why is this a topic worthy of research? This will help set the scene and point readers to the usefulness of what is to follow.

The authors thank the reviewer for this comment. As well as stating the rationale for the importance of this review, we have now detailed the importance of research on health literacy education as follows:

One key area of intervention is improving health literacy and health behaviours among adolescents which is an important public health topic given the strong links between health literacy and adult health outcomes [1] as well as generally promoting health during the life course [2].

Methods:

9. Row 148 mentions that studies were either qualitative or quantitative but the results section suggests there were 20 mixed methods studies.

The authors thank the reviewer for noting this, this has been corrected to state qualitative ‘and/or quantitative’.

10. Inclusion /exclusion: It might be helpful to consider use of a table to present the inclusion / exclusion criteria.

We have now added the inclusion/exclusion criteria to a table as suggested which we hope is now clearer.

a. In limitations, the article mentions that only studies pertaining to a “whole class or year group” were included, however this is not mentioned in the inclusion criteria.

The authors thank the reviewer for highlighting this missing detail which has now been added to the inclusion criteria.

b. Earlier the article refers to including studies that focus on health literacy and health behaviours. Point 3 (row 145) refers to health-related outcomes (health knowledge, attitudes, or behaviours). Consistency of concepts will help reader and focus study.

The authors thank the reviewer for highlighting this inconsistency. We have now used ‘health literacy and health behaviours’ consistently throughout the manuscript. 

c. Row 161, should also exclude counselling intervention as this was mentioned earlier as beyond scope of study.

The authors thank the reviewer for highlighting this, this addition has been actioned.

11. Rows 167 to 171 … Where their discrepancies between reviewers? What were these? How were they resolved?

We included the following detail in the manuscript: “Typically, disagreements arose during the initial coding process where we detailing types of mechanisms appearing in the paper, before we had consolidated the themes and confirmed the names of the mechanisms. These discrepancies were resolved through iterative discussions between EW and SD as well as with the wider team.”

12. What type of thematic analysis was used? What was the process of theme development?

The authors have now added further detail about the process of thematic analysis which details as follows:

The analysis followed aspects of the framework approach [37], primarily by creating an analytical framework to code all extracted mechanisms. After reading the included papers and extracting any relevant mechanism data into the extraction table, EW and SD made initial notes and started a set of preliminary codes. EW and SD agreed the list of codes which closely described all the mechanisms that were discussed in the included papers. This process went through several iterations through discussions between EW and SD as well as the wider author team. Once the coding framework (set of identified mechanisms) was agreed upon, EW and SD used the final framework to code all relevant mechanisms data within the extraction table. We went through an iterative process of comparing and consolidating mechanisms between papers and reduced an initial larger number of mechanisms to reach the final 10 included. 

13. Rows 192-198 I don’t believe are necessary. For transparency the MMAT could be attached as supplementary material displaying the criteria which was used to inform decision making.

The authors have removed this detail from the manuscript as suggested. 

14. Should sentence in row 180 be combined with sentences in rows 172-174.

We have now combined these sentences as suggested.

Results:

15. Referencing style changes in tables.

16. Figure 1 may be better located in methods section.

17. Rows 215-218 maybe belong better in methods section.

The authors thank the reviewer for this comment. We have chosen to keep this within the results section as this only became apparent after our analysis took place.

18. HIV/AIDS prevention. It would be helpful to not have these as interchangeable acronyms. HIV prevention would be suffice.

The authors thank the reviewer for this comment. The manuscript now only contains ‘HIV prevention’.

19. Pg 11. Did studies with theories differ in quality of mechanisms used to others without?

The authors thank the reviewer for raising this question. We did not systematically compare papers with and without theories in relation to quality of mechanisms. Quality of mechanisms was not a focus of this review, rather the review aim was to identify key mechanisms of peer education. Given that papers included in this review often lacked depth in their description of mechanisms, and did not relate them to specific outcomes, assessing quality in any systematic way would have been difficult. 

20. I have not provided extensive feedback on the themes. I see similarities across some of the themes and believe that there needs to be further analysis conducted which could considerably reduce the number of themes

---

## [Editor Report · Decision Letter 1]

4 Apr 2024

Mechanisms of school-based peer education interventions to improve young people’shealth literacy or health behaviours: a realist-informed systematic review.

PONE-D-23-18256R1

Dear Dr. Widnall,

We’re pleased to inform you that your manuscript has been judged scientifically suitable for publication and will be formally accepted for publication once it meets all outstanding technical requirements.

Kind regards,

Sogo France Matlala, PhD

Academic Editor

PLOS ONE
---

## [Editor Report · Acceptance letter]

17 May 2024

PONE-D-23-18256R1 

PLOS ONE

Dear Dr. Widnall, 

I'm pleased to inform you that your manuscript has been deemed suitable for publication in PLOS ONE. Congratulations! Your manuscript is now being handed over to our production team.

Kind regards, 

on behalf of

Professor Sogo France Matlala 

Academic Editor

PLOS ONE